# Quality of Life as Perceived by Elite Mountain Athletes in Spain

**DOI:** 10.3390/healthcare11162320

**Published:** 2023-08-17

**Authors:** Jorge Rojo-Ramos, Noelia Mayordomo-Pinilla, Antonio Castillo-Paredes, Carmen Galán-Arroyo

**Affiliations:** 1Physical Activity for Education, Performance and Health (PAEPH) Research Group, Faculty of Sports Sciences, University of Extremadura, 10003 Cáceres, Spain; nmayordo@alumnos.unex.es; 2Grupo AFySE, Investigación en Actividad Física y Salud Escolar, Escuela de Pedagogía en Educación Física, Facultad de Educación, Universidad de Las Américas, Santiago 8370040, Chile; 3Physical and Health Literacy and Health-Related Quality of Life (PHYQoL), Faculty of Sport Science, University of Extremadura, 10003 Cáceres, Spain; mamengalana@unex.es; 4Department Sport and Well-Being, School of Education, Branco Polytechnic Institute, 6000-266 Castelo Branco, Portugal

**Keywords:** elite athletes, quality of life, emotional regulation, sex, urban, rural, mountain athletes

## Abstract

Introduction: Physical activity is a great remedy to prevent diseases, as well as to keep us healthy and improve our physical, mental, and social health. One of the many benefits of physical exercise is emotional regulation, which allows us to provide an adequate response to everyday situations in addition to controlling our own emotions. High-level athletes face multifactorial stressors that can affect their quality of life. Materials and Methods: We explored the relationship between quality of life and emotional regulation using questionnaires that measure self-reported quality of life and how they cope with stressful situations in 54 mountain athletes with a mean age of 21.88 (SD = 7.88). We also investigated gender differences and demographic location in this population, as they are subjected to very high moments of stress in competition, with the risk that this modality entails. Results: Rural areas have better physical and psychological health, with higher scores on quality of life and adaptation dimensions. Women have a worse quality of life, specifically in psychological health, with worse coping mechanisms. Conclusions: It is important to design strategies that improve these mechanisms, specifically in urban areas and the female sex, to improve their emotional regulation and quality of life.

## 1. Introduction

The concept of quality of life has evolved over time, with authors disagreeing on how to define it because of its complexity. Fernández-López defined it as a set of subjective dimensions and conditions of each person, influenced by the conditions of the environment and culture, which contribute to satisfying the individual’s expectations, especially in relation to health [1]. In addition to health, other dimensions that must be satisfied include emotional well-being, interpersonal relationships, personal development, physical well-being, self-determination, social inclusion, and rights, which contribute to health status [2]. Physical activity (PA), defined as any movement performed by the musculoskeletal system that requires caloric expenditure, is one of the most useful tools for improving the quality of life of people [3]. Performing PA on a recurrent basis is defined as physical exercise, being a planned activity for the improvement of aspects of physical fitness, which is a concept that refers to aspects of each person’s health [3]. In this sense, people with average physical fitness have the ability to perform daily life activities and enjoy their leisure time without extreme exhaustion by overcoming obstacles satisfactorily, making PA a fundamental factor in maintaining quality of life [4,5]. Scientific research has determined that physical activity produces an improvement in health, helps to preserve it, reduces the incidence of diseases, and maintains risk factors at bay [6,7]. In relation to these benefits, PA also has an impact on the psychological well-being of individuals [8,9], reducing rates of anxiety, depression, and stress, among other disorders. In line with psychological well-being, emotional regulation plays an important role in mental health. Emotional regulation can be defined as the ability to control, feel, and express emotions, managing the effect of these feelings on the actions that the individual performs or how they respond to these feelings [10]. Many factors pose a problem for proper emotional regulation, such as stress, an agent that can be defined as situations that require a high demand for effort at an unexpected time, causing threatening contexts that exceed the individual’s ability to respond [11]. The scientific community has studied the capacity of physical activity as an emotional regulator, finding that it influences this aspect. In individuals who do not exercise regularly, even five-minute walking sessions are effective in improving emotional regulation and coping with emotions [12]. 

Although PA is a very good tool for the improvement and preservation of health and quality of life, those who dedicate themselves to it as elite and high-performance athletes are exposed to different risks derived from this practice. High-level athletes are those who have highly developed physical fitness and an extraordinary ability within their discipline; these athletes obtain the best results, stand out from the rest, and compete at the highest level [13]. In these types of sports, there is an important risk factor, since there is a wide range of external elements that can influence the performance of athletes, such as orientation, instability of the environment, and the high physical demand of these sports. High-performance athletes must always give the maximum of their abilities, sometimes pushing their body and mind to the limit to obtain the best possible result, increasing the risk of muscular injuries, which can even become chronic, and psychological discomfort owing to the social and media pressure they suffer [14]. To better understand the factors that produce this distress, the scientific community has investigated this field and identified elements that trigger it. Among these elements, there are stressors of different natures, being organizational, mental, and physical: sports at this level contain organizational stressors and normalization of harmful habits, such as eating disorders, injuries, harassment, and isolation [15,16,17]. All these factors add up and worsen the mental health of athletes, facilitating the development of psychological disorders, such as anxiety, depression, eating disorders, and burnout syndrome [18,19]. By developing strategies to control their emotions during the rigors of competition, athletes can reduce the influence of emotional fluctuations on their decision-making processes [20]. The significance of emotional regulation becomes even more pronounced within the elite athlete population, as research indicates that these individuals often possess a tendency towards catastrophizing events, perceiving them in an excessively negative light. The importance of this element is even more important in the elite since the literature has found that these individuals tend to have a more catastrophic view of events [21]. Regarding emotional regulation between sexes, studies show that the female sex has worse emotional regulation, although this may be due to the fact that they tend to have better communication skills than men, who tend not to share these feelings, repressing them, and not dealing with them [22,23].

In addition, training and competition involve a high physical load for these individuals, generating a high rate of muscular tension and stress, which, in a context where biopsychosocial conditions are unfavorable, could generate an injury [24]. In addition, these high loads of effort and training can generate pain that, over time, can become chronic [25], causing this population to resort to analgesic drugs to tolerate this pain [24]. These factors, both physical and psychological, interfere with the perceived quality of life of elite athletes, with worse emotional regulation that could interfere with performance.

After contextualizing the state of research in this field, it is of interest to delve into the possible relationships between quality of life and emotional regulation in this specific population, looking for correlations between these two variables, since the way they manage their emotions and feelings can have a significant impact on their quality of life. After reviewing the literature on the aspects of quality of life and coping mechanisms, it is hypothesized that women will report a poorer quality of life than men, having worse coping mechanisms. It is also hypothesized that athletes with better coping mechanisms will have a better quality of life.

## 2. Materials and Methods

### 2.1. Participants

The study is a descriptive cross-sectional design. Convenience sampling-based non-probabilistic sampling was used to choose the sample [26]. It may be considered that the sample’s gender distribution, 68.5% males and 31.5% females (n = 54), is balanced. Participants had to meet inclusion requirements such as being certified as technification, high-performance, or high-level athletes in any of the disciplines considered as mountain sports by the Spanish Federation of Mountaineering and Climbing (mountaineering, trekking, canyoning, skiing, Nordic walking, hiking, or climbing). Other factors were selected to describe the sample (Table 1), including the athletes’ sport modality, educational attainment, and geographic location (rural areas were classified as those with fewer than 20,000 residents). The Body Mass Index, calculated from the weight and height values provided by the participants, had a mean value of 20.99 (SD = 2.64) while the mean age was 21.78 years (SD = 7.88). The minimum age was 14 years and the maximum age was 40 years.

### 2.2. Procedure

Digital means were used to acquire the data, and the methodology was used to generate self-administered questionnaires; this form facilitates data collection for later use, saving time and money [27]. The possibility of carrying them out remotely is their main advantage; in this case, the questionnaire was created using the online tool Google Forms and accessed through the link (https://forms.gle/UF9RwQLy685ZH6tZ9, accessed on 1 September 2021). The three items in the questionnaire—two of which were the instruments and one a guide for correct understanding—were completely anonymous. It took an average of three minutes to complete each task. The entire dataset was collected between October 2022 and March 2023. The questionnaire was sent to the Mountain and Climbing Federations of the different regions of Spain so that they could send it to high-level, technical, or high-performance athletes, and the use of social networks and email was also employed.

### 2.3. Instruments

First, a survey was created with six sociodemographic questions (sex, age, demographic location, height, weight, and type of sport). The following formula was used to determine BMI: Height in meters/weight in kilos^2^. 

The Cognitive Emotion Regulation Questionnaire (CERQ) was used to evaluate emotional control [28]. The 36 items that make up the CERQ test measure 9 cognitive-emotional coping mechanisms for handling challenging circumstances and events. This instrument is based on a 5-point Likert scale, where 1 equals “sometimes” and 5 equals “always”.

The questionnaire is composed of the following dimensions: (1) Self-blame (“I feel that I am to blame for what happened”); (2) Acceptance (“I think I have to accept what has happened”); (3) Rumination (“I often think about how I feel in relation to what has happened to me”); (4) Positive focus (“I think of something more pleasant than what has happened to me”); (5) Planning (“I think about what is the best thing I could do”); (6) Positive reassessment (“I think I can learn something from the situation”); (7) Perspective taking (“I think it could have been much worse”); (8) Catastrophism (“I often think that what has happened to me is much worse than what has happened to other people”), and (9) Blame others (“It seems to me that others are to blame for what happened”). These 9 factors can also be divided into disadaptive strategies (self-blame, rumination, catastrophism, and blame others) and adaptive strategies (acceptance, positive focus, planning, positive reassessment, and perspective taking). Internal consistency for the various subscales, as shown by Cronbach’s alpha, ranges from 0.68 (blame others) to 0.83 (rumination). Additionally, the internal consistency of the various subscales ranged from 0.62 (catastrophism) to 0.83 (positive focus) in the Spanish version for teenagers [29].

Lastly, using the WHOQOL-BREF [30], the self-perceived quality of life was examined. The 26 items on the instrument were split into four categories: (1) Physical health (7 items), which covers mobility, daily activities, functional ability, energy, pain, and sleep; (2) Psychological health (6 items), which covers mentality, learning ability, memory concentration, religion, and state of mind; (3) Social relationships (3 items), which supplements information on interpersonal relationships, social support, and sex life, and (4) Environmental health (8 items), which addresses concerns with access to financial resources, safety, health, and social services, the physical living environment, possibilities for learning new skills, leisure activities, the general environment (noise, air pollution, etc.), and transportation. A Likert-type response scale is employed to score each individual questionnaire item from 1 to 5, and the results for each dimension are then converted into a scale from 1 to 100. Studies such as [31] that analyzed the reliability of this instrument, reported reliability values in physical health as 0.77, psychological health as 0.77, social relations as 0.76, and environment as 0.72, demonstrating appropriate reliability.

### 2.4. Statistical Analysis

The analysis of the data collected was performed with the Statistical Package for Social Sciences (SPSS) version 23.0 for MAC (IBM Corporation, Armonk, NY, USA).

Three negative WHOQOL-BREF items were reversed before the analysis to converge the analysis domain. Since a sample size of approximately 50 participants was acquired, the distribution of the data was examined to check if the presumption of normality was met using the Shapiro–Wilk test with the aim to determine the kind of statistical tests to be employed [32]. It was decided to utilize nonparametric statistical tests since this test revealed that the presumption was false.

Then, the Mann–Whitney U test was utilized, with a significance level of *p* < 0.05, to examine the variations in scores for each of the dimensions based on sex or demographic location. Additionally, the Spearman’s Rho test was performed to assess the degree of correlation between each of the dimensions and either sex or demographic location. The parameters specified by Mondragón-Barrera [33] were considered for the interpretation of this statistic: between 0.01 and 0.10, a coefficient denotes the presence of a low correlation; between 0.11 and 0.50, a medium degree of correlation; between 0.51 and 0.75, a strong correlation; between 0.76 and 0.90, a high correlation, and above 0.91, a perfect correlation.

Finally, Cronbach’s alpha was utilized to evaluate the instrument’s reliability. Consistency rates between 0.60 and 0.70 can be regarded as acceptable, whereas values between 0.70 and 0.90 can be regarded as satisfactory, according to Nunnally and Bernstein [34].

## 3. Results

According to demographic location and sex, Table 2 displays descriptive statistics based on mean and standard deviation for each of the CERQ dimensions. When examining differences between groups, the Mann–Whitney U test provided the statistical significance.

With the exception of self-blame and catastrophism, demographic location revealed the highest scores in rural settings for the majority of variables. Acceptance (*p* = 0.01), planning (*p* = 0.01), positive reassessment (*p* = 0.01), and adaptation strategies (*p* = 0.01)—all of which favored rural settings—were also found to differ in statistically significant forms. In terms of gender, females outperformed males on most dimensions of the CERQ, except for positive focus, catastrophism and blame others. However, statistically significant differences in gender were only obtained for rumination (*p* = 0.03) and planning (*p* = 0.03).

Similarly, Table 3 shows the scores and differences obtained in each of the dimensions of the WHOQOL-BREF instrument. In regard to demographic situation, significant differences were observed in both the psychological (*p* = 0.01) and physical health (*p* = 0.01) dimensions, with higher scores in rural settings except for the latter dimension (environmental health). When the focus is on gender, males show higher scores than their female counterparts in all dimensions, although significant differences were found only in psychological health (*p* = 0.01). 

Table 4 shows the correlation coefficients obtained in Spearman’s Rho test, carried out in order to explore the associations between emotional regulation and quality of life both at a general level and according to gender and demographic location. At the general level, the only dimension of the CERQ that showed significance when associated with quality of life was self-blame, reporting a medium and inverse correlation. In terms of demographic location, again self-culpability seems to be the only factor that is related to quality of life, with this relationship being average, inverse, and with a higher coefficient in rural environments. Sex exposes the same trend as before, with self-blame as the only dimension that stands out in its relationship with quality of life. In this case, the values for women are strong, inverse, and significant, while for men they have the same characteristics except for being a mean coefficient.

The Cronbach’s alpha values, mean, standard deviation, skewness, and kurtosis provided for each of the CERQ and WHOQOL-BREF dimensions are shown in Table 5 as the final results. Regarding reliability, except for the social relations dimension, acceptable values were found (between 0.70 and 0.90), even with a small sample size.

## 4. Discussion

The main objective of this study was to explore the possible relationship between the quality of life of high-performance mountain athletes and their ability to regulate their emotions. To achieve this objective, the CERQ questionnaire for emotional regulation capacity and the WHOQOL-BREF questionnaire for self-perception of the quality of life of these athletes were administered. 

Regarding the results obtained in the descriptive analyses of the CERQ dimensions on the differences between sexes and the environment in which the athlete was located, significant differences were observed. In terms of the demographic location variable, athletes from rural environments scored higher than those living in a city in terms of acceptance, planning, positive reassessment, and adaptive strategies. These results indicate that the rural environment tends to accept what has happened positively, to plan the best alternatives to a problem, and to take advantage of these unfavorable situations to learn and make the best of them than those who live in urban environments, being more resilient and coping better with these stressful situations. The environment in which athletes find themselves has a great impact on the way they deal with situations that cause stress and discomfort, influencing the strategies they choose and the extent to which these situations affect them [35]. However, no scientific studies have been found to support or contradict these results, as the environment in which athletes live remains an unexplored field. In relation to gender, significant differences were found in the rumination and planning dimensions, with women scoring higher. This means that women face these situations by planning the actions they could take and also thinking about how this event makes them feel on a recurrent basis. In line with the results obtained in this study, scientific literature obtained higher scores in rumination and planning but also in catastrophism, although no significant differences were found in this study [21,36,37]. This finding may be due to the greater tendency of females to express their emotions and develop them, as the male gender, as a general rule, is not in the habit of doing so. Along the same lines, it is clear that these emotions of vulnerability are not well regarded socially by men, since other types of emotions, such as strength, are associated with them [22,23]. In this case, only the first hypothesis is partially accepted, since women report a lower quality of life in the psychological aspect but do not report worse coping mechanisms in all dimensions.In relation to the information obtained from the application of the WHOQOL-BREF questionnaire, information on the perceived quality of life of each athlete was obtained through the four dimensions that make up this tool. Significant differences were found in two of these dimensions in the demographic location variable, with athletes living in rural areas scoring higher. These dimensions were physical health and psychological health, showing a higher quality of life related to physical and psychological well-being. Studies that have explored the differences in mental health depending on the environment in which they live by applying this tool show disparate results, in which some find that the urban environment has a better quality of psychological life [38] and others obtain results in line with those presented in this work [39]. However, the social context of the place where the research population resides is of key importance, since the perception of quality of life is influenced by health, socio-political factors, and the degree of economic development of each country, and the difference between urban and rural environments differs depending on this context [40,41]. Regarding physical health, studies that apply this tool to explore the differences between rural and urban environments do not find significant differences, although it is expected that the rural environment will obtain better results because of the presence of green environments with restorative properties [42]. However, further research is required in this area to establish relationships. On the other hand, males obtained higher scores in all dimensions of this tool, finding significant differences in the psychological dimension. This finding coincides with that found in another study in which the male sex dominated this dimension, explaining that women have a greater tendency to have poorer mental health and develop more depressive and anxiety disorders than men [43,44]. In the exploration of the correlations between the dimensions of the CERQ questionnaire and perceived quality of life, significant correlations were found only in the self-blame dimension, which is negative and of medium strength. This correlation indicates that the lower the score obtained in this dimension, the higher the quality of life perceived by the athletes; in line with this result, the study by Manju [45] expressed correlations of the same type and degree in this dimension, together with acceptance, rumination, and catastrophizing. In this sense, the second hypothesis put forward at the beginning of the document is accepted.Related to the variables studied in this paper, significant correlations appear in the rural environment, with a medium degree of inverse character, indicating that this dimension has more influence on this population. As a general rule, there is a greater religious tendency in rural areas than in larger cities, and a study that applied the WHOQOL-BREF tool to measure religious beliefs and coping methods found that a score below the mean in quality of life correlated with maladaptive strategies including self-blame, especially in those who were religious [46]. Regarding sex, women had a strong and inverse correlation in this aspect, explaining that self-blame has a great weight in maladaptive strategies in the quality of life of elite sportswomen. In contrast to this result, a previous study found no correlation or sex differences [45]. However, another study found a greater propensity of women to self-blame due to a greater tendency to focus on negative feelings and emotions and on how these affect them, looking for a culprit [47]. In addition, the greater likelihood of women suffering from depressive disorders and society’s expectations of both genders makes them more vulnerable and prone to this type of behavior [48,49].

### 4.1. Practical Applications

With the analyses obtained, it is possible to draw a profile of the most vulnerable athletes with the worst quality of life to implement programs to improve this perception and thus reduce the incidence of mental disorders that this type of population tends to suffer. The most vulnerable profile is the elite athlete who lives in an urban area and is female, with a high propensity to self-blame; however, those who live in rural areas have a greater tendency to use self-blame and have a poorer quality of life. In this sense, the implementation of programs that have an impact on improving the coping mechanisms of athletes should focus on improving this dimension.

### 4.2. Limitations and Future Lines

Although interesting evidence was obtained, the results should be interpreted with caution. The data were collected using an electronic questionnaire to save materials and facilitate data management, however, this type of questionnaire has some disadvantages [50]. Having segmented the sample into groups, the sample is even smaller, so this section should be considered as a limitation. It is important to note that comparisons by age group could not be made because although the age range was 14 to 40 years, there was a small number of athletes in the same range. In addition, cultural factors were not considered, so it is possible that this type of factor influenced the way in which athletes deal with stressful situations; therefore, the results should be interpreted with caution when comparing them with other countries. For future studies, it would be interesting to extend the sample to other technification or high-performance centers, involving other regions to increase the range of sociocultural factors that may affect the behavior of the sample. 

## 5. Conclusions

This study aimed to explore the perceived quality of life of elite mountain athletes using the WHOQOL-BREF questionnaire in relation to the dimensions of the CERQ questionnaire, looking for correlations between them and exploring the existing differences between the environment in which they reside and the sex of these athletes. The results showed that, in general, athletes from rural areas scored higher than those from urban areas, finding significant differences in adaptive responses, revealing that these athletes have a more positive way of dealing with disadvantageous events. These results show that elite rural athletes have a higher physical and psychological quality of life, probably due to positive coping strategies. Together with the negative correlation in the self-blame dimension, it is important to use these results for the design of strategies to improve their ways of coping with obstacles, particularly focusing on this dimension. 

Regarding sex, it has been shown that women have a worse perceived quality of life, possibly caused by their management and coping with obstacles, since, in general, the results are higher for negative items than for positive ones. In addition, their mental health also suffers more than men’s as the results show, although this may be because men tend to hide these types of emotions as they can be seen as a weakness and because socially, a stronger behavior is imposed on them. It is essential to apply strategies to improve these dimensions, fundamentally focusing on self-blame, where the negative correlation is strong and could substantially improve quality of life.

## Figures and Tables

**Table 1 healthcare-11-02320-t001:** Sociodemographic characteristics of the sample (N = 54).

Variable	Categories	N	%
Sex	Men	37	68.5
	Women	17	31.5
Education level	Secondary education	28	51.9
	Professional training	8	14.8
	University	10	18.5
	Master’s or Doctorate	8	14.8
Demographic location	Rural environment	23	42.6
	Urban environment	31	57.4
Athlete’s condition	Technification	23	42.6
	High performance	28	51.9
	High level	3	5.6

N: number; %: percentage.

**Table 2 healthcare-11-02320-t002:** Comparing descriptive analysis with CERQ dimensions.

Dimension	Demographic Location			Sex		
	Rural	Urban		Male	Female	
	M (SD)	M (SD)	*p*	M (SD)	M (SD)	*p*
Self-blame	3.09 (0.77)	3.25 (0.81)	0.44	3.09 (0.80)	3.38 (0.76)	0.21
Acceptance	12.26 (1.84)	10.56 (2.49)	0.01 *	11.02 (2.18)	11.85 (2.72)	0.26
Rumination	11.09 (2.41)	10.70 (2.89)	0.74	10.30 (2.55)	12.10 (2.61)	0.03 *
Positive focus	8.63 (2.73)	8.26 (2.45)	0.67	8.48 (2.42)	8.27 (2.89)	0.70
Planning	13.67 (1.97)	11.17 (2.42)	0.01 *	11.77 (2.37)	13.26 (2.68)	0.03 *
Perspective taking	11.48 (2.76)	10.86 (2.14)	0.43	10.97 (2.17)	11.45 (2.93)	0.41
Catastrophism	7.56 (3.13)	7.83 (2.57)	0.68	7.80 (2.73)	7.17 (2.96)	0.29
Blame others	7.13 (2.31)	7.08 (2.43)	0.81	7.35 (2.42)	6.50 (2.19)	0.30
Adaptive strategies	11.99 (1.47)	10.44 (1.64)	0.01 *	10.88 (1.62)	11.58 (1.94)	0.16
Disadaptive strategies	7.22 (1.44)	7.16 (1.72)	0.94	7.13 (1.65)	7.30 (1.50)	0.68

M = Mean; SD = standard deviation. Each score obtained is based on a Likert scale (1–5). * *p* is significant < 0.05.

**Table 3 healthcare-11-02320-t003:** Descriptive information and variations in each WHOQOL-BREF dimension depending on demographic location and sex.

Dimension	Demographic Location			Sex		
	Rural	Urban		Male	Female	
	M (SD)	M (SD)	*p*	M (SD)	M (SD)	*p*
Physical health	4.31 (0.42)	3.90 (0.56)	0.01 *	4.15 (0.52)	3.93 (0.58)	0.13
Psychological health	4.19 (0.53)	3.73 (0.73)	0.01 *	4.09 (0.64)	3.56 (0.65)	0.01 *
Social relationships	3.82 (0.79)	3.48 (0.83)	0.142	3.71 (0.75)	3.45 (0.97)	0.33
Environmental health	3.76 (0.58)	3.92 (0.57)	0.30	3.95 (0.05)	3.63 (0.65)	0.15

M = Mean; SD = standard deviation. Each score obtained is based on a Likert scale (1–5). * *p* is significant < 0.05.

**Table 4 healthcare-11-02320-t004:** Correlation between each of the CERQ dimensions and self-perceived quality of life as a function of gender and demographic location.

Dimension	WHOQOL-BREF *ρ* (*p*)	Demographic Location		Sex	
		Rural	Urban	Male	Female
		*ρ* (*p*)	*ρ* (*p*)	*ρ* (*p*)	*ρ* (*p*)
Self-blame	−0.36 (<0.01 *)	−0.41 (0.04) *	−0.22 (0.22)	−0.22 (0.17)	−0.62 (0.01) *
Acceptance	0.03 (0.85)	0.11 (0.59)	−0.09 (0.61)	0.01 (0.92)	0.03 (0.89)
Rumination	−0.20 (0.14)	−0.29 (0.17)	−0.15 (0.41)	−0.14 (0.41)	−0.15 (0.56)
Positive focus	0.02 (0.87)	0.20 (0.35)	−0.14 (0.44)	−0.13 (0.43)	0.20 (0.43)
Planning	0.06 (0.64)	−0.27 (0.20)	0.14 (0.43)	0.10 (0.54)	0.34 (0.17)
Positive reassessment	0.17 (0.21)	0.01 (0.94)	0.07 (0.69)	0.12 (0.47)	0.44 (0.07)
Perspective taking	0.09 (0.50)	−0.01 (0.97)	0.15 (0.39)	0.03 (0.82)	0.24 (0.35)
Catastrophism	−0.17 (0.21)	−0.20 (0.36)	−0.17 (0.34)	−0.25 (0.14)	−0.18 (0.49)
Blame others	0.07 (0.64)	0.21 (0.32)	0.03 (0.86)	−0.07 (0.68)	0.35 (0.17)
Adaptive strategies	0.12 (0.37)	0.08 (0.69)	−0.00 (0.99)	0.08 (0.62)	0.34 (0.18)
Disadaptive strategies	−0.18 (0.18)	−0.22 (0.30)	−0.18 (0.32)	−0.18 (0.29)	−0.06 (0.80)

Each score obtained is based on a Likert scale (1–5). The correlation was significant at * *p* < 0.05.

**Table 5 healthcare-11-02320-t005:** Reliability values for the dimensions of the scales.

Instrument	Cronbach’s Alpha	M (SD)	Asymmetry	Kurtosis
CERQ				
1. Self-blame	0.70	3.18 (0.79)	0.25	−0.87
2. Acceptance	0.72	11.28 (2.37)	−0.42	0.66
3. Rumination	0.77	10.87 (2.68)	−0.08	−0.73
4. Positive focus	0.72	8.42 (2.55)	0.51	−0.70
5. Planning	0.72	12.24 (2.55)	−0.23	−0.79
6. Positive reassessment	0.85	12.43 (3.02)	−0.23	−0.79
7. Perspective taking	0.71	11.12 (2.42)	0.09	−0.25
8. Catastrophism	0.70	7.60 (2.79)	0.15	−0.90
9. Blame others	0.74	7.10 (2.36)	0.22	−0.49
10. Adaptive strategies	0.70	11.10 (1.74)	−0.42	−0.32
11. Disadaptive strategies	0.70	7.19 (1.59)	−0.12	−0.82
WHOQOL-BREF				
1. Physical health	0.70	4.08 (0.54)	−0.58	−0.30
2. Psychological health	0.74	3.92 (0.69)	−0.91	0.25
3. Social relationships	0.60	3.62 (0.83)	−0.43	−0.07
4. Environmental health	0.70	3.85 (0.58)	−0.54	−0.37

M = Mean; SD = standard deviation.

## Data Availability

The datasets are available through the corresponding author upon reasonable request.

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
