# Peer review of "Quality of Life as Perceived by Elite Mountain Athletes in Spain"

_healthcare, 2023, doi:10.3390/healthcare11162320_

Round 1
Reviewer 1 Report
This is an interesting study. It is easy to follow, and the results look good. Also, the discussion is based on their results and adopts some previous knowledge. I believe that it is worth publishing to show our readers.
There are some minor points to make a revised paper to submit.
1. In the abstract, please explain why the authors chose elite mountain athletes.
2. L22-24 In the abstract, please delete how you obtain your data by statistical analysis.
3. In the abstract, you may add some information about your results and your thoughts and consideration derived from your results.
4. L108-100 You may need to delete this part.
5. L117 You may need space after a period.
6. In the section of Statistical Analysis, please add information if you used any application to get your results.
This is my end of suggestion to make a revised manuscript.
Overall, the text looks good but may need to be checked by native English speakers.
Author Response
REVIEWER 1
This is an interesting study. It is easy to adhere to and the findings seem good. Moreover, the discussion is based on your findings and adopts some previous knowledge. I think it is worth publishing it to show it to our readers. There are some minor points to make a revised paper to present.
Author's reply: First of all, thank you very much for your time and input. Thanks to this, the manuscript has been greatly improved.
- In the abstract, please explain why the authors chose elite mountain athletes.
Author's reply: This population was chosen specifically because of its relevance to the data obtained, since, in the demographic characteristics, it is most practiced in high performance mountaineering. In addition, it is important to take into account the risk and therefore stress involved in the sport. We have included this in the summary. Thank you very much for your comment.
- L22-24 In the abstract, please delete how data are obtained by statistical analysis.
Author's reply: Dear reviewer, thank you very much for your appreciation. We have removed this part of the abstract to make it easier to read.
- In the abstract, you can add some information about your findings and your reflections and considerations derived from your results.
Author's reply: Thank you very much for this comment. We have added a couple of lines on practical applications derived from the findings.
- L108-100 You may need to delete this part.
Author's reply: Thank you very much for this suggestion. The paragraph was duplicated and we have deleted it.
- L117 You may need space after a full stop.
Author's reply: OK, we have added it. Thank you very much for pointing this out.
- En la sección de Análisis Estadístico, por favor añada información si utilizó alguna aplicación para obtener sus resultados.
Author's reply: Thank you, reviewer. The analysis of the collected data was performed with the Statistical Package for Social Sciences (SPSS) version 23.0 for MAC (IBM Corporation, Armonk, NY, USA). We have added it to the manuscript.
This concludes my suggestion for a revised manuscript.
Author's reply: Thanks a lot for your suggestions.
Reviewer 2 Report
Admissions to such an important study seem too low. It is helpful to find different points of view, especially from the point of view of mental health research
The section on limitations should also refer to limitations related to the size or structure of the sample and the method of its selection.
Can the country of study - e.g. lifestyle in Spain - influence the results? Do you expect high volatility, e.g. in other European countries?
To phrase: Regarding gender, it has been shown that women have a worse managed quality of life, possibly caused by management and coping with obstacles, since, in general, the results are higher for negative items than for positive ones. In addition, their mental health also suffers more than that of men"
Try to discuss this with research that suggests that men are less likely to admit to mental health crises. Perhaps this conclusion is not true.
Author Response
REVIEWER 2
Author's reply: First of all, thank you very much for your time and input. Thanks to this, the manuscript has been greatly improved.
Admissions to such an important study seem too low. It is useful to find different points of view, especially from the point of view of mental health research. The section on limitations should also refer to limitations related to the size or structure of the sample and the method of its selection. ¿Can the country of study - e.g. lifestyle in Spain - influence the findings? Is high volatility expected, e.g. in other European countries?
Author's reply: Thank you very much for this note. We have included in the limitations section the cultural factor, especially in mental health. It may have influenced the way athletes deal with their emotions.
To paraphrase: In terms of gender, women have been shown to have a worse managed quality of life, possibly caused by managing and coping with obstacles, as, in general, the findings are higher on negative items than on positive ones. In addition, their mental health also suffers more than men's." Try to argue this with the research that suggests that men are less likely to admit to mental health crises. Perhaps this conclusion is not true.
Author's reply: Thank you very much for this note. We have rewritten this statement, as it was not entirely correct taking into account social and cultural factors; however, we have added information to support this new statement.
Reviewer 3 Report
The manuscript is well structured and functional to the research objectives, in all its parts, and it is grammatically well written; however, it would be desirable to redesign the abstract by including the contents in sections (intro, materials and methods, results, conclusions) and to include 1 or 2 summary tables in the discussions, so as to facilitate reading. The study design also involves the use of only one psychometric instrument, whereas in similar studies at least 2 tests are used; however, the result obtained is valuable.
Author Response
REVIEWER 3
The manuscript is well structured and functional to the research objectives, in all its parts, and is grammatically well written; however, it would be desirable:
- Redesign the abstract by including the contents in sections (introduction, materials and methods, findings, conclusions).
- Author's reply: Thank you very much for your comment. We have restructured the abstract to include these parts so that you can identify which part it is about.
- Include 1 or 2 summary tables in the discussions, to facilitate reading.
- Author's reply: Thanks for your suggestion. The discussion has been structured in sections to make it more visible.
3.The study design also involves the use of only one psychometric instrument, whereas in similar studies at least 2 tests are used; however, the result obtained is valuable.
- Author's reply: Thanks for your appreciation. As we will adhere to this line of work, we have taken your feedback into consideration and will incorporate at least 2 tests in the following studies.
Reviewer 4 Report
Dear authors,
Thank you very much for your interesting article. However, I consider there are several aspects that need to be changed
ABSTRACT
- Include number of subjects, age, and standard deviation
- Keywords: removing the acronyms CERQ, WHOQOL-BREF, is not the most appropriate. Even in the Abstract I would remove it
INTRODUCTION
- What were considered high-level athletes? It is important to describe what this means. Were the chosen athletes from a specific sport? In this case, include the description of the sport, limiting factors... more context about said sport
- The study hypothesis should be included
MATERIALS AND METHODS
- Include study design (Design 2.1)
- Participants: What sports did you play? How old were the participants? The standard deviation is 7.88 out of 21.78 (were there participants aged 13-14?). It would be interesting to differentiate between athletes according to age range
- Was the sample size calculated to ensure that it was representative? It seems few participants
- Procedure: the link to Google forms that was used must be included. Likewise, in the case of being minors, was any informed consent given?
- Instruments: the reliability of the WHOQOL-BREF questionnaire must be included
- Statistical analysis: it is indicated that the required sample was 50 participants. How was this sample size calculated? Was any procedure used to filter out the atypical cases of the study? (for example, having randomly answered the tests)
RESULTS
- Due to the small sample, when it is divided by demographic location or sex, the size is very small, this must be taken into account especially as a limitation of the research
- The results should be described after the table with the statistical significance values.
- Table 5: I consider that these values ​​could be previously included by making a table where the mean, standard deviation, skewness, kurtosis and correlations between the variables appear, as an initial descriptive study.
DISCUSSION
- I think it should be restructured, indicating the research hypotheses at the end of the introduction, and responding to them in the discussion section.
Author Response
REVIEWER 4
Dear authors, thank you very much for your interesting article. However, I feel that there are several aspects that need to be modified:
ABSTRACT
- Include number of subjects, age and standard deviation.
- Keywords: removing the acronyms CERQ, WHOQOL-BREF, is not the most appropriate. Even in the Abstract I would remove it
Author's reply: Dear reviewer, thank you very much for these appreciations. We have removed the acronyms from both the abstract and the keywords, including the number and age of the sample to improve the quality of the abstract.
INTRODUCTION
- What are high-level athletes? It is important to describe what this means. Were the athletes chosen from a specific sport? If so, include a description of the sport, limiting factors... more context about the sport.
- The hypothesis of the study should be included
Author's reply: Thank you very much for your feedback. We have restructured this part in the introduction to give more context about this sport, as well as adding the hypothesis.
MATERIALS AND METHODS
- Include study design (Design 2.1)
Author's reply: Thank you for your appreciation. The study is a descriptive cross-sectional design and this information has been incorporated into the method.
- Participants: What sports did they play? How old were the participants? The standard deviation is 7.88 out of 21.78 (were there participants aged 13-14?). It would be interesting to differentiate the athletes according to age range.
Author's reply: Thank you very much, the athletes had to practice one of the sports modalities included in the catalogue of the Spanish Federation of Mountaineering and Climbing: (Alpinism, mountaineering, trekking, canyoning, skiing, Nordic walking, hiking or climbing) This has been specified in the inclusion criteria section. The minimum age was 14 years and the maximum age was 40 years. We have NOT included this detail in the description of the sample, but if you think so, we will do so without any problem. We are very grateful for your suggestion to differentiate the participants by age groups, but as the sample was so small due to the characteristics that the athletes had to meet, we have found that there are age ranges with very few participants. However, as we are extending this study to other modalities, we will adhere to your suggestion, which we found very interesting.
- Was the sample size calculated to ensure representativeness? It seems that there were too few participants.
Author's reply: The sample was not calculated as we could not obtain a detailed number of athletes classified as high performance, technification and high level athletes as the Spanish Mountaineering Federation does not have this register. One of the reasons for the delay in responding to your comments was mainly due to the fact that we have tried to contact all the regional federations but this census does not exist.
- Procedure: the link to the Google forms that were used should be included. Also, in the case of minors, was any informed consent given?
Author's reply: Thank you very much for your appreciation, we have included the link to the form in the procedure section. In the case of minors, participants had to have parental consent in order to participate in the study, by claiming on the same form that they had such consent.
- Instruments: the reliability of the WHOQOL-BREF questionnaire should be included.
Author's reply: Thank you very much, the reliability values have been included in the Whoqol-Bref section.
- Statistical analysis: it is stated that the required sample was 50 participants, how was the sample size calculated, was any procedure used to filter out atypical cases from the study (e.g. randomly responding to the tests), and was the sample size calculated?
Author's reply: Thank you very much, after consulting the Spanish federation of mountaineering and climbing, there is no register that groups athletes classified as high performance, high level and technification athletes. For this reason it was not possible to perform the sample calculation. With regard to the atypical cases of the study and to prevent desirability in the choice of answers, it was decided not to take into consideration those questionnaires in which all the answers to all the items were polarised; however, this was not the case in any of the surveys.
RESULTS
- Due to the small sample size, when divided by demographic location or gender, the sample size is very small, this should be taken into account especially as a limitation of the research.
Author's reply: Thank you very much for your appreciation. We have incorporated this as a limitation of the research.
- The findings should be described after the table with statistical significance values.
Author's reply: Thank you very much, the significance values have been included when describing the findings after the tables.
- Table 5: I think that these values could be included previously by making a table where the mean, standard deviation, skewness, kurtosis and correlations between variables are shown, as an initial descriptive study.
Author's reply: Thank you very much for your appreciation. We have included in the same table the values of the mean, SD, skewness and kurtosis. The correlations between variables are already included in the other tables according to the objectives of the study. Thank you very much and we found your suggestion very interesting.
DISCUSSION
- I think it should be restructured, indicating the research hypotheses at the end of the introduction, and responding to them in the discussion section
Authors' response: thank you very much for your input. We have adhered to your indications and the discussion has been restructured again.
Round 2
Reviewer 4 Report
Thank you to the authors for your effort to improve the manuscript.
I consider it could be published. However I would like that the “problem” with sample size and the different age will be included in limitation section. On the other hand, the age range should be included in sample section
Congratulations,
Author Response
Reviewer 4:
Thanks to the authors for their efforts to improve the manuscript.
I consider it publishable. However, I would like the "problem" of sample size and age difference to be included in the limitations section. On the other hand, the age range should be included in the sample section.
Authors' response: Thank you very much for your consideration. We have incorporated all your suggestions and the article has been greatly improved.